Effects of deuterium oxide on cell growth and vesicle speed in RBL-2H3 cells

Kalkur Roshni S. 1
Ballast Andrew C. 1 2
Triplett Ashley R. 1
Spendier Kathrin 1 2 kspendie@uccs.edu
1 BioFrontiers Center, University of Colorado at Colorado Springs , Colorado Springs, CO , USA
2 Department of Physics and Energy Science, University of Colorado at Colorado Springs , Colorado Springs, CO , USA
Sun Shao-Chen
Electronic publication date: 2014 Sep 2
Publication date: 2014
Volume: 2
Electronic Location ID: e553
Received 2014 Jun 18; Accepted 2014 Aug 10
Copyright: © 2014 Kalkur et al.
Copyright year: 2014
Copyright holder: Kalkur et al.
License: This is an open access article distributed under the terms of the Creative Commons Attribution License, which permits unrestricted use, distribution, reproduction and adaptation in any medium and for any purpose provided that it is properly attributed. For attribution, the original author(s), title, publication source (PeerJ) and either DOI or URL of the article must be cited.
License URL: https://creativecommons.org/licenses/by/4.0/

Keywords: RBL-2H3 cells, Flow cytometry, Deuterium oxide, Single-particle tracking, TIRF microscopy, Microtubule-dependent vesicle transport

Funding: This work was supported by the University of Colorado, Colorado Springs BioFrontiers Center. The funders had no role in study design, data collection and analysis, decision to publish, or preparation of the manuscript.

==============================
For the first time we show the effects of deuterium oxide on cell growth and vesicle transport in rat basophilic leukemia (RBL-2H3) cells. RBL-2H3 cells cultured with 15 moles/L deuterium showed decreased cell growth which was attributed to cells not doubling their DNA content. Experimental observations also showed an increase in vesicle speed for cells cultured in deuterium oxide. This increase in vesicle speed was not observed in deuterium oxide cultures treated with a microtubule-destabilizing drug, suggesting that deuterium oxide affects microtubule-dependent vesicle transport.

Introduction

Roughly 70% of Earth’s surface and animal bodies are made out of water (H2O). Very few, if any, biological systems or reactions will function without water and one may conclude that the properties of H2O are essential for life on Earth. In recent years research has indicated that water plays an active role in how biomolecules recognize and bind to each other (Ball, 2011). For example, in a biological system when a protein binds its ligand, associates with another protein, or folds into its functional form, the surrounding solvent must get out of the way. How water may act as a versatile intermediary and facilitator during these processes is still under investigation (Ball, 2011).

To study the effect of water (H2O), one must find ways to change the properties of H2O. This can be accomplished by substituting hydrogen (H) by its heavier deuterium (D) isotope resulting in deuterium oxide (D2O), also known as heavy water or heavy-hydrogen water. All naturally-occurring water contains approximately 150 parts per million D and therefore, D2O may be essential for some life forms (Lewis, 1934). Deuterium contains one proton and one neutron and bonds to oxygen (O) more strongly than H (one proton and no neutron) in H2O. This results in small differences in the length of the covalent H–O-bonds and the angles between them, thus making D2O roughly 11% denser and 25% more viscous than H2O at 20 °C (Hardy & Cottington, 1949). Due to the natural occurrence of D2O and differences in chemical structure and physical properties compared to H2O, researchers have used D2O to study the effects of water on biomolecules and cells. Studies include effects of D2O on tobacco seed growth (Lewis, 1934), IgE-Mediated histamine release from human leukocytes (Gillespie & Lichtenstein, 1972), actin filament velocities (Chaen et al., 2001), protein flexibility (Cioni & Strambini, 2002), human pancreatic tumor cells (Hartmann et al., 2005), phospholipid membranes (Beranova et al., 2012), kinesin-1 gliding assay (Maloney, Herskowitz & Koch, 2014), and stabilization of tubulin as observed previously in microtubule gliding assays (Panda et al., 2000) and in biochemical experiments on isolated tubulin proteins from beef brain (Houston et al., 1974) and goat brain (Das et al., 2008). Microtubules are polarized polymers of α/β tubulin heterodimers and undergo alternating phases of growth and shrinkage with sudden transitions between the two (Bartolini et al., 2005). Microtubules are responsible for a wide variety of vital cellular functions such as the formation of a bipolar spindle at mitosis. (Mitosis is the process, in the cell cycle, by which a cell duplicates into two genetically identical daughter cells.) However, it is still unknown how stabilization of tubulin, i.e., the stability of microtubules, due to D2O affects microtubule-dependent vesicle transport in cell cultures.

In this paper we use the RBL-2H3 cell line, a typical mast cell model system (Thomas, Feder & Webb, 1992; Posner et al., 1995; Carroll-Portillo et al., 2010; Spendier et al., 2010), to start investigating this question. Mast cells are immune cells that originate from the bone marrow and circulate in an immature form in the body until they settle in tissue and mucosal surfaces, where they mature. When mast cells are activated by allergens, they cause allergic responses and protect the body from parasitic infection. A key player in this activation process is the IgE-receptor complex (IgE–FcϵRI). Allergens or multivalent ligands crosslink IgE-receptor complexes on the cell surface, leading to receptor phosphorylation (Metzger, 1992; Galli, Tsai & Piliponsky, 2008). A subsequent signaling cascade results in degranulation, i.e., fusion of secretory granules with the plasma membrane (exocytosis) to release a number of inflammatory mediators including histamine and serotonin (Galli, Tsai & Piliponsky, 2008). One focus of the paper is to investigate the effect of D2O on the transport speed of these secretory granules (secretory vesicles) that are known to be transported in a direct and microtubule-dependent manner towards the cell plasma membrane for exocytosis (Smith et al., 2003).

To our knowledge, the presented investigations have not been performed previously on RBL-2H3 cells. In this paper, we study the effects of D2O on RBL-2H3 cell growth and vesicle transport. For this first investigation, one specific concentration of 15 moles/L D2O was chosen. This concentration was previously reported to significantly increase histamine release in human leukocytes (Gillespie & Lichtenstein, 1972). Experiments outlined below indicate that this concentration of D2O slows RBL-2H3 cell growth and causes an increase in vesicle speed. Specifically, the data suggests an increase in vesicle speed for microtubule-dependent transport.

Methods

Cells

The RBL-2H3 cell line was purchased from ATCC. RBL-2H3 cells were maintained in minimal essential medium (MEM) supplemented with 10% fetal bovine serum (FBS), 1% Penicillin Streptomycin (Pen-Strep), and 1% L-glutamine (L-glut). For the experiment cells were grown under 5% CO2 atmosphere in 6-well dishes for up to five days with media made from MEM powder containing 0 moles/L or 15 moles/L D2O. Anti-DNP IgE was purchased from Sigma-Aldrich. Before microscopy, cells were IgE primed by incubation with 0.5 mg/mL of IgE overnight. After the addition of cells to the microscope imaging chamber, cells were stimulated with DNP-conjugated BSA (DNP25-BSA) at 2 µg/mL for up to 30 min. For colchicine drug treatment, cells were treated with 100 µM colchicine for 45 min (Smith et al., 2003). Trypan blue stained cells were used to count viable (alive) and dead cells on a hemacytometer. The presented studies on this cell line have been approved by the University of Colorado at Colorado Springs Institutional Review Board, approval number IBC 13-001.

Supported lipid bilayers

Prior to use, microscope glass cover slips were cleaned of organic residues with a mixture of sulfuric acid and hydrogen peroxide (piranha solution). Supported lipid bilayers (Carroll-Portillo et al., 2010; Spendier et al., 2010) were made by spontaneous liposome fusion (Werner et al., 2009). Lipids, obtained from Avanti, were dissolved in chloroform, dried under air flow, and then placed under a vacuum for 1 h to remove traces of oxygen. The lipid film was then suspended in PBS to 1.3 mM and sonicated for 5 min using a probe sonicator in an ice bath. Laterally mobile bilayers were formed from 1-palmitoyl-2-oleoyl-sn-glycero-3-phosphocholine (POPC) and 5 mol% N-dinitrophenyl-aminocaproyl phosphatidylethanolamine (DNP-Cap PE) on piranha-cleaned cover glass for 15 min on a slide warmer at 37 °C. Each bilayer coated coverslip was kept water immersed during transfer to the imaging chamber. Prior to adding cells to the bilayer, the chamber was flushed with Hanks’s buffered saline solution.

DiI labeling

For fluorescent imaging of lipid membrane vesicles, approximately 3 × 105 cells were suspended in 1 mL of Hanks’s buffered saline solution and incubated with 2 µL Vibrant DiI (Life Technologies, Grans Islan, NY) at 2 mM concentration for 2 min. After incubation, the cells were washed at least three times in Hanks’s buffered saline solution.

Flow cytometry measurements

RBL-2H3 cell cycle was analyzed by quantification of DNA content with a Beckman Coulter Cytomics FC 500 MPL Flow Cytometer using forward scatter (FS), side scatter (SS), and fluorescence 3 (FL3, 620 nm Band Pass Filter) channels. The brightness of the propidium iodide (PI) fluorescent dye was used to evaluate DNA content. A 488 nm laser beam was focused onto the flowing stream at a fixed point, illuminating the cells as they passed through. To prepare cell samples, cells were first trypsinated and then washed with MEM of the appropriate D2O concentration. Cells were fixed and PI-stained by re-suspending cell pellets in 250 µL of MEM, 250 µL of ethanol and 100 µL of PI at 0.1 g/L. PI-stained cell samples were refrigerated overnight and analyzed the following day. Samples were run 100 s or until the count exceeded 10,000 cells. For analysis, the single cell population was first gated using FS vs. SS to remove obvious cell debris and non-cellular elements. This gate was then applied to FS vs. FL3 to reduce doublets, i.e., two cells stuck together that may register as a single cell. These two gates were then combined and applied to give the final PI histogram plots.

Fluorescent imaging

DiI stained RBL-2H3 cells were imaged using a total internal reflection fluorescence (TIRF) microscope. Cell samples were maintained at 37 °C using an objective heater. In TIRF microscopy, cells were allowed to settle onto a fluid lipid POPC bilayer with 5 mol% DNP lipid or on piranha-cleaned glass under gravity. Objective-based TIRF microscopy was performed with an S-TIRF module (Spectral Applied Research, Canada) attached to a Leica DMI3000 B inverted microscope with a 100 × and 1.47 N.A. oil immersion objective using a 561 nm laser (Coherent Inc.) excitation. A 1.5 × lens was also added to the excitation beam path resulting in a final magnification of 150 ×. The penetration depth of the evanescent wave for 561 nm excitation was calculated to be 200 nm. A 600/50 nm single-band bandpass filter (Chroma) was used to collect fluorescence. TIRF images were collected with an EMCCD camera (Evolve Delta; Photometrics) operated by Micro-Manager (Stuurman, Amodaj & Vale, 2007).

Single-vesicle tracking

Image processing was conducted in MATLAB (MathWorks, Inc., Natick, MA), in conjunction with the DIPImage image processing toolbox (Delft University of Technology). Image backgrounds were averaged and subtracted to reduce noise. Fluorescent particles of size within the microscope resolution limit (Gaussian variance of 200 nm) were identified as vesicles. Vesicle coordinates were identified in each frame by a direct Gaussian fit algorithm and a cost function was employed to link coordinates together into trajectories (Andrews et al., 2008). Trajectories were histogrammed by average hop speed for each vesicle trajectory. Hop speed was calculated as a function of distance/frame, where each frame represents a 20 ms time interval. Any distinct fluorescent clusters of low speed, defined as less than 10−2 µm/s, were removed to separate vesicles from stationary cellular structures. The localization uncertainty for individual trajectory positions was within 100 nm.

Results and Discussion

The effect of deuterium oxide on RBL-2H3 cell growth and viability

It is known that D2O can reduce cell growth in human pancreatic tumor cells (Hartmann et al., 2005). To confirm that a similar trend is observed in RBL-2H3 cells, cell growth and cell viability was monitored over a period of 5 days. RBL-2H3 cells were cultured in 6 well plates in cell media containing 0 moles/L D2O or 15 moles/L D2O. The plates were seeded at the same time to ensure similar initial cell seeding concentration for each well. Cells were allowed to divide for 24 h before the first measurement. Figure 1A shows that when cells were cultured in 15 moles/L D2O (open circles) a 7-fold decrease in RBL-2H3 cell growth was observed after five days compared to cells cultured in 0 moles/L D2O (closed circles). Cell viability also dropped by 20% for cells cultured in D2O (open circles) as shown in Fig. 1B. These results are qualitatively consistent with studies on human pancreatic tumor cells (Hartmann et al., 2005).

Figure 1 The effect of deuterium oxide on RBL-2H3 cell (A) growth and (B) viability.

Closed circles represent cells cultured with 0 moles/L D2O and open circles cells cultured with 15 moles/L D2O. Each data point is computed from the mean of three individual experiments and the error bar represents the standard deviation. Cells were seeded on day zero.

The observed reduction in cell growth and viability may be caused by differences in pH between the two growing conditions (Lardner, 2001). For cell media containing 0 moles/L D2O and 15 moles/L D2O, the average pH was 7.58 ± 0.29 and 7.48 ± 0.12, respectively during a 5 day period. The error is represented by the standard deviation. No statistically significant difference was observed and therefore we conclude that changes in pH did not cause the observed decrease in cell growth.

Finally, to start investigating the effect of D2O on RBL-2H3 cell cycle, flow cytometry measurements were performed to monitor the RBL-2H3 cell cycle by measuring DNA content (Darzynkiewicz, Robinson & Crissman, 1994). The cell cycle has four distinct phases that can be recognized in a proliferating cell population: the G1-(growth phase), S-(DNA synthesis phase), G2-(growth phase and preparation for mitosis) and M-phase (mitosis). Here, PI was used as a DNA probe to investigate the cell cycle, in particular the G0–G1-, S- and G2–M-phases. Each of these phases can be identified by DNA content. G0–G1 is diploid and has a normal complement of DNA. In S-phase, DNA synthesis occurs and in the G2–M-phase, twice the amount of DNA is found in the cells than in the G0–G1 phase. Therefore, a typical DNA content frequency histogram of proliferating (multiplying) cells shows two clearly separated peaks, a G0–G1-peak and a lower G2–M-peak separated by the cell subpopulation in the S-phase.

Figure 2 depicts DNA content frequency histograms for RBL-2H3 cells in untreated cultures (Fig. 2A) and in cultures treated with D2O (Fig. 2B). The presented data was smoothed using a moving average and the area underneath the curve was normalized to unity. For 0 moles/L D2O treated cells, the G2–M-phase (black arrow) is recognizable in one-day (black line in Fig. 2A) and four-day (red line in Fig. 2A) old cultures. In other 0 moles/L D2O cell cultures (data not shown) the G2–M peak was more pronounced. In one-day old D2O treated cultures, the G2–M-phase can also be identified (black line in Fig. 2B). The number of cells entering the G2–M-phase in four-day old D2O cultures decreases, as shown by a weakened G2–M-phase and an increased height of the G0–G1 peak (red line in Fig. 2B). The loss of the G2–M peak reflects that RBL-2H3 cells have not doubled their DNA content. This loss of DNA content may be caused by cells failing to enter mitosis and/or is a sign that cells are in the early stages of apoptosis. Both mechanisms were reported previously in human pancreatic tumor cells in which D2O induced apoptosis in PANC-1 and AsPC-1 cells and arrested PANC-1 and BxPC-3 cells in the G2–M-phase of the cell cycle (Hartmann et al., 2005). We also note that the G0–G1 peaks for both D2O treated and untreated cultures experience a shift of approximately the same magnitude to the left within four days. Such shifts can be attributed to daily fluctuations in the flow cytometry excitation and detection system.

Figure 2 The effect of deuterium oxide on RBL-2H3 cell cycle as shown by propidium iodide intensity (FL3) histogram plots.

Cell counts were normalized to unit area. Cells grown (A) in media containing 0 moles/L D2O and (B) in media containing 15 moles/L D2O. The black lines represent one-day old cultures and the red lines represent four-day old cultures. The arrows indicates position of the G2–M-cell cycle phase.

DiI-stained RBL-2H3 cells for vesicle tracking

As pointed out above, one possible mechanism for failure to double DNA is cell arrest near the G2–M-phase of the cell cycle. G2–M-cell cycle arrest can be associated with problems in the mitotic spindle structure (Holy, 2002). The mitotic spindle is a structure composed of microtubules which segregates chromosomes into the daughter cells during mitosis. Besides structural support of microtubules, they also act as highways within a cell for trafficking a wide variety of cargo by molecular motors. Here we focus on secretory granules as cargo that are transported by molecular motors (kinesins) towards the cell plasma membrane for exocytosis during mast cell activation via IgE-FcϵRI crosslinking (Smith et al., 2003). It has been reported previously that stable microtubules result in increased Kinesin-1 motor speeds, since motors that move along dynamic microtubules could rapidly run off the end of the microtubule (Cai et al., 2009). Experiments also showed that D2O has a microtubule stabilization effect in microtubule gliding assays (Panda et al., 2000) and on isolated tubulin proteins (Houston et al., 1974; Das et al., 2008). Therefore vesicles that are transported by molecular motors in a microtubule-dependent manner should experience different transport speeds in untreated and D2O treated cultures. Specifically, the mean vesicle hop speed of tracked secretory vesicles in D2O treated cultures is expected to increase. To mark these secretory vesicles for the studies presented here, diI was used as fluorescent lipophilic membrane stain (Anantharam et al., 2010). DiI is weakly fluorescent until incorporated into lipid membranes. It labels all cell membrane barriers, including the cell plasma membrane and membrane-bound organelles such as the nucleus, endoplasmic reticulum, golgi apparatus, lysosomes, endosomes, and mitochondria (see Fig. S1).

TIRF microscopy was employed to reduce background fluorescence from the cell plasma membrane and fluorescence from membrane-bound organelles. In TIRF microscopy, the excitation laser beam is totally internally reflected at the glass-water interface and only an evanescent wave traveling parallel to the interface penetrates into the sample. The evanescent wave decays rapidly with a perpendicular distance from the interface, and selectively excites the sample within a distance of about 200 nm from the surface. Besides reduction of background fluorescence, the advantage of using TIRF microscopy is selective excitation of vesicles that are within the evanescent field. In mast cells, membrane-bound granules are approximately 300–400 nm in diameter (Martynova et al., 2005). Therefore in the experiments outlined below, vesicles were identified as fluorescent particles of size within the microscope resolution limit. Since secretory granules are transported in a direct manner towards the cell plasma membrane for exocytosis, TIRF microscopy makes it possible to track these vesicles.

To test the ability of diI to label secretory vesicles, diI-stained, IgE-loaded RBL-2H3 cells cultured in 0 moles/L D2O were allowed to settle onto glass over a 5 min period, and then stimulated with multivalent DNP25-BSA for 30 min. Cells were imaged before the addition of DNP25-BSA and during the incubation period using TIRF microscopy at 20 frames/s. Movie S1 clearly shows sustained vesicle motion 26 min after DNP25-BSA stimulation (right panel in top row of Movie S1). Some vesicle motion, albeit slower, was also observed in unstimulated RBL-2H3 cells which is consistent with previous work by Smith et al. (2003). To shorten cell-settling time to less than 1 min and achieve good cell-substrate contact required for TIRF imaging, cells were allowed to settle onto a supported lipid bilayer with 5 mol% DNP-Cap PE for the single-vesicle tracking experiments outlined below. One of the authors showed previously that RBL-2H3 cells in contact with liganded-bilayers composed of 5 mol% DNP-Cap PE exhibit degranulation levels similar to un-stimulated cells (Carroll-Portillo et al., 2010).

Effect of D2O on vesicle motion in RBL-2H3 cells

As pointed out above, diI labels all membrane barriers including vesicles that are not transported on microtubules. However, if a large number of vesicles in DNP25-BSA activated cells are tracked within the cell-substrate contact area, a significant population of these vesicles is expected to be secretory, i.e., vesicles that are transported on microtubules for exocytosis. Therefore, over 1,500 individual vesicles were tracked using TIRF microscopy. For each vesicle-trajectory, the mean hop speed was calculated. The average of these mean vesicle hop speeds for cells cultured in 0 moles/L D2O was 3.6 ± 0.1 µm/s. This average speed was significantly slower than the average of 8.5 ± 0.4 µm/s computed from the mean vesicle hop speeds for cells cultured in 15 moles/L D2O. This difference was statistically significant as determined by the Student’s t-test with a p-value of less than 0.05.

To test whether some of the diI-labeled vesicles tracked with TIRF microscopy are secretory, a known microtubule destabilizing drug called colchicine was used which is expected to slow down secretory vesicle transport (Smith et al., 2003). For cells cultured in 15 moles/L D2O the average of mean vesicle hop speeds for colchicine treated cells decreased significantly to 2.4 ± 0.3 µm/s as expected for microtubule-dependent vesicle transport. However, for cells cultured in 0 moles/L D2O the average of all mean hop speeds for colchicine treated cells did not significantly change and was measured to be 4.2 ± 0.3 µm/s. Possible reasons for not measuring a decrease in vesicle hop speed for colchicine treated cells cultured in 0 moles/L D2O are as follows. Firstly, the camera projects a three-dimensional path of a moving object onto a two-dimensional plane. Therefore, the component of vesicle velocity parallel to the substrate is not the full vesicle velocity vector. If only a small parallel component of an outward (from cell center towards cell-substrate contact zone) directed vesicle velocity vector is seen, there is no reason to insist that the parallel component by itself be large enough to generate a statistically detectable difference. 15 moles/L D2O treated cells appear to have larger vesicle velocities than 0 moles/L D2O treated cultures. Cells with larger overall vesicle velocities also have larger parallel components, and those larger parallel components show a statistically significant change with the application of colchicine. Secondly, the reported average hop speed includes both microtubule-dependent and microtubule-independent vesicle transport. Therefore, the colchicine treatment may have a different effect on these two subpopulations. Microtubules extend from the nucleus to cell surface. The disruption of microtubules results in a loss of these structures that may alter the transport of non-secretory vesicles. Once this structure is disrupted, non-secretory vesicles may be able to move more freely (diffuse faster), which results in an increase of the average hop speed for this specific population. This change in vesicle speed for non-secretory vesicles can counteract or dominate the expected decrease in microtubule-dependent vesicle speeds. By culturing cells in D2O, it appears that microtubule-stabilization is the dominant factor that increases the overall speed of vesicle transport. Finally, vesicles with low hop speed of less than 10−2 µm/s were classified as stationary structures and therefore were not used to calculate the average of mean vesicle hop speeds. This proportion of stationary vesicles is expected to increase after treating cells with colchicine. Indeed a ten-fold increase from 0.043 to 0.418 and from 0.065 to 0.580 in 0 moles/L D2O and 15 moles/L D2O cultures was observed, respectively. This analysis shows a clear effect of colchicine on vesicle traffic in both D2O and non-D2O treated RBL-2H3 cultures.

Figure 3 shows four box plots of mean hop speeds obtained from analyzing over 1,500 individual vesicle trajectories. Going from left to right, the first and second plot represent mean vesicle hop speeds in cells cultured in 0 moles/L D2O without colchicine (Fig. 3, H2O) and with colchicine (Fig. 3, H2O with colchicine) treatment, respectively. The third and fourth plot represent mean vesicle hop speeds in cells cultured in 15 moles/L D2O without colchicine (Fig. 3, D2O) and with colchicine (Fig. 3, D2O with colchicine) treatment, respectively. The median for mean vesicle hop speeds for cells cultured in 0 moles/L D2O was 2.3 µm/s. This median speed was lower than the median of 5.3 µm/s computed from the mean vesicle hop speeds for cells cultured in 15 moles/L D2O. For cells cultured in 15 moles/L D2O the median for colchicine treated cells decreased to 1.4 µm/s as expected. However, for cells cultured in 0 moles/L D2O the median of mean vesicle hop speeds for colchicine treated cells increased to 3.6 µm/s . This comparison also indicates that colchicine treatment may have different effects on microtubule-dependent and microtubule-independent vesicle transport. It is likely that the outliers of the distributions (red crosses in Fig. 3) for colchicine untreated cultures represent mean vesicle hop speeds for microtubule-dependent transport. This claim is supported by the following. Firstly, cargo transport by kinesin motors along microtubules can be faster than microtubule-independent (diffusive) protein transport (Koon, Koh & Chiam, 2014). Secondly, these outliers decrease significantly for colchicine treated cells cultured in both 0 moles/L and 15 moles/L D2O. However, to fully investigate deuterium oxide’s microtubule stabilization effect on secretory vesicle transport, future experiments that clearly identify secretory vesicles are warranted. For example, one can follow Smith et al. (2003) and transfect RBL-2H3 cells with a green fluorescent protein-Fas ligand fusion protein (GFP-FasL) to study the transport of GFP-labeled secretory vesicles in more detail. Such investigations were beyond the scope of the presented study. Nevertheless, the reported results can be used as a springboard for future investigations.

Figure 3 Box plots of mean vesicle hop speeds.

Going from left to right plots represent mean vesicle hop speeds for cells cultured in 0 moles/L D2O without (H2O) and with (H2O with colchicine) colchicine and cells cultured in 15 moles/L D2O without (D2O) and with (D2O with colchicine) colchicine. The red crosses represent outliers and the red lines the median for each distribution. Going from left to right, the median for each distribution was 2.3 µm/s, 3.6 µm/s, 5.3 µm/s, and 1.4 µm/s, respectively.

Conclusions

The primary goal of our investigations above was to study the unknown effects of D2O on RBL-2H3 cell growth and vesicle transport. A seven-fold decrease in cell growth was observed in five-day old cultures for cells grown in 15 moles/L D2O compared to untreated cells. Using flow cytometry, the decrease in cell proliferation was attributed to cells not sufficiently doubling their DNA content. Using TIRF microscopy in conjunction with single-vesicle tracking in antigen-stimulated and diI-labeled RBL-2H3 cells, the average vesicle hop speed increased significantly in D2O cultures and decreased after treating cells with colchicine, a microtubule-destabilizing drug. These results indicate that a subpopulation of tracked vesicles were transported along microtubules for vesicle exocytosis. This observation together with previous reports on D2O’s microtubule stabilization effect in microtubule gliding assays (Panda et al., 2000) and on isolated tubulin proteins (Houston et al., 1974; Das et al., 2008), suggest that the observed increase in vesicle hop speed for D2O treated RBL-2H3 cultures may be due to stabilization of tubulin. Since G2–M-cell cycle arrest can be associated with problems in mitotic spindle structure (Holy, 2002), increased stability of microtubules may be the cause for the observed decrease in RBL-2H3 cell proliferation. It remains to be investigated whether D2O causes RBL-2H3 cells not to enter mitosis and/or induces apoptosis.

Supplemental Information

Movie S1 Movie S1 shows diI-stained, IgE-loaded RBL-2H3 cells that settled onto glass under gravity. Cells were unstimulated or stimulated with multivalent DNP25-BSA for more than 26 min and imaged with TIRF microscopy at 20 frames/s. Time t = 0 min is approximately 5 min after initial cell-substrate contact. Top row: cell before DNP25-BSA stimulation (left panel), after approximately 20 min DNP25-BSA stimulation (middle panel), and after approximately 26 min DNP25-BSA stimulation (right panel). Bottom row: resting (unstimulated) cell in contact with the glass substrate at t = 0 min (left panel), at t = 20 min (middle panel), and at t = 26 min (right panel).

Click here for additional data file.

Figure S1 To confirm incorporation of diI into the cell plasma membrane and membrane-bound organelles, diI-stained RBL-2H3 cells were imaged using a confocal microscope. Cell samples were maintained at 37 °C using an objective heater. For confocal imaging, a Leica TCS SP5 confocal laser scanning microscope with a 63 × oil immersion objective was used. diI-labeled cells were excited with 543 nm laser light in confocal microscopy. Appropriate filter settings were used to collect fluorescence. Figure S1 depicts a confocal z-stack montage of a diI-labeled RBL-2H3 cell and shows that diI labels cell membrane barriers, including the cell plasma membrane and membrane-bound organelles such as the nucleus, lysosomes, and endosomes. This figure shows a confocal z-stack montage of a diI-labeled RBL-2H3. All panels correspond to individual confocal z-slices starting with the z-slice of the cell in contact with the glass substrate, panel A, and going up to the top of the cell, panel X. The z-step size was set to 692 nm. Scale bar represents 10 µm.

Click here for additional data file.

Supplemental Information 3 Cell count and viability data for Fig. 1

Data in form of XLSX files

file name description:

Alive and Dead Cell Count: data for Fig. 1A

Cell Viability: data for Fig. 1B.

Click here for additional data file.

Supplemental Information 4 Flow Cytometry raw data for Fig. 2

LMD files are the Flow Cytometry raw data for Fig. 2.PDF files are overview pages showing analyzed data with the applied gatesFile name description:dayoneH2O: one-day old culture with 0 mol/L deuterium oxide dayfourH2O: four-day old culture with 0 mol/L deuterium oxide dayoneD2O: one-day old culture with 15 mol/L deuterium oxide dayfourH2O: four-day old culture with 15 mol/L deuterium oxide.

Click here for additional data file.

The authors would like to thank Dr. Yuriy Garbovskiy for cell culturing, William Townend for useful discussions and help with flow cytometry measurements and Dr. Nathan Zameroski for useful content related discussion.

Additional Information and Declarations

Competing Interests

Author Contributions

Ethics

Data Deposition

The authors declare that there were no competing financial, professional or personal interests that might have influenced the performance or presentation of the work described in this manuscript.

Roshni S. Kalkur and Andrew C. Ballast conceived and designed the experiments, performed the experiments, analyzed the data, prepared figures and/or tables, reviewed drafts of the paper, these authors contributed equally to this work.

Ashley R. Triplett conceived and designed the experiments, performed the experiments, analyzed the data, reviewed drafts of the paper.

Kathrin Spendier conceived and designed the experiments, analyzed the data, contributed reagents/materials/analysis tools, wrote the paper, prepared figures and/or tables, reviewed drafts of the paper.

The following information was supplied relating to ethical approvals (i.e., approving body and any reference numbers):

University of Colorado at Colorado Springs Institutional Review Board, IBC 13-001.

The following information was supplied regarding the deposition of related data:

TIRF data uploaded to FigShare:

DOI 10.6084/m9.figshare.1059555

DOI 10.6084/m9.figshare.1060175

DOI 10.6084/m9.figshare.1060377

DOI 10.6084/m9.figshare.1060390

DOI 10.6084/m9.figshare.1060416

DOI 10.6084/m9.figshare.1060428

DOI 10.6084/m9.figshare.1060436

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
