# Peer review of "Effects of deuterium oxide on cell growth and vesicle speed in RBL-2H3 cells"

_PeerJ, doi:10.7717/peerj.553_

## Round 0.1 · original submission · Major Revisions

Both reviewers raised concerns about the data interpretation like microtubule dependent-vesicle transport, and additional experiment is needed to confirm whether the cells enter mitosis or is going to apoptosis.

Reviewer 1 ·

Basic reporting

Major issue:The basic reporting is flawed in a couple of ways.
1. The authors selectively ignore important aspects of their data.
A. The cell sorting data indicate that the cells in D2O do not increase their DNA content. The authors conclude that the cells are blocked "near" G2/M. The data do show that for D2O treated cells, there is a loss of the G2/M peak - merely reflecting that cells have not doubled their DNA content. However, the entire DNA staining histogram for day 4 in D2O, is left-shifted from the day one cells. This is ignored. PI staining only shows DNA content such that the notion of blockage "near" G2/M is unjustified. All they can say is that DNA was not replicated. On the other hand, the loss of DNA content deserves discussion.One possibility is that the cells are in the early stages of apoptosis. This would not be detected as cell death by Trypan Blue staining.
B. In Figure 3, the authors compare the effect of colchicine in H2O and D2O. Colchicine decreases the hop speed of D2O treated cells whereas, if anything, the speed is increases when cells grown in H2O are treated with colchicine. The authors conclude that the colchicine effect in D2O treated cells shows that transport/secretion is microtubule dependent. However, they do not comment on the failure of colchicine to alter the same parameters in cells grown in H2O. So, the readers asks, does this mean that vesicle transport in normal media is not microtubule dependent? The microtubule-dependence of vesicle transport is documented in the literature they cite (Smith et al., 2003). So, one wonders if there is a flaw in the experiment design/results such that they do not detect an effect of colchicine under normal circumstances.

2. They authors make assertions by way of explanation that have no associated rationale or justification.
A. On P10 "A possible mechanism for the observed cell cycle arrest in RBL-2H3 cells
cultured in D2O is microtubule stabilization. In this case, vesicles that are
transported in a microtubule-dependent manner should experience different
transport speeds in untreated and D2O treated cultures. As shown by
Smith et al. (2003) during mast cell activation via IgE-FceRI crosslinking,
secretory granules are transported in a direct and microtubule-dependent
manner towards the cell plasma membrane for exocytosis".

The question here is, why should microtubule stabilization lead to differences in transport speeds? Please explain this to the reader.


Minor issues:
Trackjectories is misspelled. It seems they meant “trajectories”.
page 3 “affect” microtubule-dependent … should be “affects”
p5 “quantization” of DNA content. Is this right?

Experimental design

The experimental design seems appropriate, However, some additional data would have greatly improved the paper. It should have been relatively easy to determine
A. If cells failed to enter mitosis
B. If cells were undergoing apoptosis.

Validity of the findings

Questions on the findings are related to inconsistencies in the presentation presented above.

Reviewer 2 ·

Basic reporting

This study was designed to study the effects of Deuterium Oxide on cell cycle and vesicle speed in RBL-2H3 cells.

Experimental design

The experimental design is fine.

Validity of the findings

The article shows that Deuterium Oxide would decrease cell growth and most cells stopped at G2-M-phase of the cell cycle; however, Deuterium Oxide increased vesicle hop speed. The authors concluded that the increase in vesicle hop speed for D2O treated RBL-2H3 cells was due to stabilization of tubulin because G2-M-cell cycle arrested increased stability of microtubules.

Additional comments

Although some very interesting data are presented, this manuscript cannot be accepted in the present form for publication in this Journal, and a content revision is suggested.

For the Figure 3, I cannot tell what’s for mean value for each treatment? Box with red line is the mean value? If yes, any difference between four groups? This result is very important for conclusion that “deuterium oxide affected microtubule-dependent vesicle transport”. The authors should address it clearly.

---

## Round 0.2 · Minor Revisions

The authors still need to recheck their results and solve the concern raised by Reviewer 1 "why colchicine does not affect vesicle traffic in non-D2O treated cells".

Reviewer 1 ·

Basic reporting

The manuscript is improved but needs some clarification

Experimental design

OK but specific staining of the secretory granules would have helped in the interpretation of the data.

Validity of the findings

There is concern about the colchicine results but otherwise they appear to be sound.

Additional comments

The manuscript by Kalkur et al is significantly improved. The one sticking point is whether the explanation for why colchicine does not affect vesicle traffic in non-D2O treated cells really makes sense. This is perhaps tied to the way they counted and classified vesicles.

The MS text says, “ Here we focus on secretory granules as cargo that are transported by molecular motors (kinesins) towards the cell plasma membrane for exocytosis during mast
cell activation via IgE-FceRI crosslinking (Smith et al., 2003)”. The mast cell histamine-containing vesicles are relatively large (300-400nm according to the authors) compared to many other types of vesicles. These vesicles could have been distinguished by their metachromatic staining with acridine orange. Yet the criterion for designating something as a vesicle was, “Fluorescent particles of size within the microscope resolution limit were identified as vesicles”.

The term “within the resolution limit” needs some clarification. One can see fluorescent objects that are below the limit of resolution. Of course, in this case, they will appear to be bigger than they really are but this does not mean they are resolved. So, the question here is how is the term “within the resolution limit” being used? This needs clarification. Was there a certain size cutoff?

Difference colchicine results for normal and D2O-treated cells
The question here is, if there was a statistically large enough sample of secretory granules in the case of D2O-treated cells to observe a difference in speed, why would this not have showed up in the non-D2O-treated cells also. Furthermore many other types of vesicles are transported along microtubules as well. The result is at least surprising.

Reviewer 2 ·

Basic reporting

The authors have addressed the concerns raised by my review. So it can be accepted for publication.

Experimental design

Good

Validity of the findings

Good

Additional comments

The authors have addressed the concerns raised by my review. So it can be accepted for publication.

---

## Round 0.3 · accepted · Accept

The authors addressed the concerns raised by the reviewer 1.